# Factors Affecting the Resilience of New Nurses in Their Working Environment

**DOI:** 10.3390/ijerph19095158

**Published:** 2022-04-24

**Authors:** Keunsook Park, Aeri Jang

**Affiliations:** 1Department of Nursing, Chonnam National University Hospital, Jebong-ro, Dong-gu, Gwangju 61469, Korea; pks01200@naver.com; 2Department of Nursing, Nambu University, Nambudae-gil, Gwangsan-gu, Gwangju 62271, Korea

**Keywords:** new nurses, resilience, work environment, regression analysis

## Abstract

Resilience and working environment are variables that affect turnover. However, how these two variables impact each other is unclear. This study is a descriptive correlation study to identify the factors of the nursing working environment affecting the resilience of new nurses in general tertiary hospitals. This study was conducted by convenience sampling of 233 new nurses with less than 1 year of working experience. Data were collected through questionnaires from 20 to 27 October 2019, and analyzed using correlation analysis and stepwise multiple regression analysis. The results show that the work environment impacts the resilience of new nurses by 30.2%. Specifically, the following qualities of a work environment were found to affect new nurses’ resilience, including “nursing foundation for quality of care”, “nurse participation in hospital affairs”, “nurse manager ability, leadership, and support of nurses”, “collegial nurse–physician relations”, and “staffing and resource adequacy”. These findings imply that a satisfactory working environment improves new nurses’ resilience and reduces their intent to leave their workplace. Further studies are needed to elucidate this relationship, especially considering the ever-changing work environments.

## 1. Introduction

According to the National Healthcare Retention & RN Staffing 2019 Report, hospitals incur an average additional cost of USD 328,400 for every 1% change in nurse turnover rate [1]. The report also notes that hospitals take an average of 3 months or more to refill the position vacated because of nurse turnover. In particular, it was found that 17.5% of new nurses working in hospitals resigned within one year, 33.5% within two years, and 43% within three years [2]. Therefore, lowering the turnover rate of new nurses is crucial in stabilizing the medical system’s operations and decreasing the budget, time, and effort hospitals spend on additional staffing.

Nurses’ resilience has recently received much attention as an essential factor for nurses’ professional success and workplace stressor management [3]. Resilience is the ability to cope with stressful situations and adapt to changing environments. It is also referred to as a “dynamic process encompassing positive adaptation within the context of significant adversity” [4]. In an integrative review of how nurses’ resilience affects their experiences, patient nursing, employers, peer support, job satisfaction, and turnover intention were seen to have the most significant influence on nurses’ resilience [5]. The same review also found that conflict, shift work, a work environment that promotes work–life balance, the feeling of being cared for, poor relationships with direct supervisors, and professional status impact nurses’ resilience. Therefore, resilience is a critical factor in the shortage and retainment of professional nurses [6].

A positive work environment is suggested as an important strategy in 10 of 22 studies [7]. The nurse’s work environment is not limited to its physical aspects, but also includes the work policies recognized by nurses, the environment’s conduciveness for effective nursing care, and the social environment involving interactions between individuals [8]. In addition, the nursing practice environment and new nurses’ resilience affect transition shock, affecting turnover intention [9]. Overall, research suggests that work environment and resilience are crucial variables in turnover intention.

Several in-depth studies have been done recently on how the work environment affects nurse resilience. One revealed that the resilience of new nurses affects turnover intention and that the hospital working environment mediated this [9]. There have also been studies on the work environment and resilience of emergency room [10] and intensive care unit [11] nurses, which identified work environment and resilience as factors affecting the new nurses’ transition shock [12]. However, an integrative review found that information on how work environment factors affect nurses’ resilience is still lacking [5]. This finding was proven given the difficulty in finding more studies relevant to this research.

This study aimed to identify the factors of the nursing working environment affecting new nurses’ resilience, particularly in general tertiary hospitals. This study will provide the basis for improving the working environment to help new nurses adapt to clinical settings by increasing their resilience. The specific objectives for this are as follows. This was done by identifying the differences in resilience according to the subjects’ general characteristics and the work environment. The correlation between the subjects’ work environment and their resilience and the factors affecting the subjects’ resilience were also identified.

## 2. Materials and Methods

### 2.1. Study Design and Participants

The subjects of this descriptive research were convenient sampling for nurses working in university hospitals with more than 500 beds in Korea. The study subjects were limited to new nurses who had been working in the hospital for 2–11 months. The number of subjects required was calculated for multiple regression analysis using G^*^Power 3.1.1 (Heinrich-Heine-Universität Düsseldorf, Düsseldorf, Germany). Specifically, the following information was used to calculate this number, 0.15 median effect size, 0.05 significance level, power of 0.7, and 22 predictors (4 work environment subregions and 18 general characteristics), resulting in 230 nurses needed for this study. A total of 250 new nurses were selected considering the dropout rate, and 233 were chosen as the final study subjects, as 17 responded insufficiently.

### 2.2. Measures

The questionnaire used in data collection checked for several measures, including the subject’s general characteristics, the nursing work environment, and the subject’s resilience. The specific measures are detailed below.

The subjects’ general characteristics were investigated by dividing them into demographic characteristics and work-related characteristics that can be considered factors that influence adaptability. The study followed the nursing work environment developed by Lake [13] and verified its reliability and validity using the Korean Version of Practice Environment Scale of Nursing Work Index (K-PES-NWI) [14]. The questionnaire incorporated 9 items on nurse participation in hospital affairs, 9 on the nursing foundation for quality of care, 4 on nurse manager ability, leadership, and support of nurses, 4 on staffing and resource adequacy, and 29 on collegial nurse–physician relations in 5 areas of relations (3 items). Each item was measured using a 4-point Likert scale, with a higher score implying a positive perception of the work environment. The reliability, at the time of development and study completion, using Chronbach’s α, was 0.93.

For resilience, the Korean version of the Connor-Davidson Resilience Scale (CD-RISC) [15] was adapted from Baek et al.’s [16] study with the approval of the authors. Thus, the questionnaire included 9 items on measuring hardness, 8 on measuring persistence, 4 on measuring optimism, 2 on measuring support, and 2 on spirituality. Each item was measured using a 5-point Likert scale, with a higher score meaning higher resilience. Using Chronbach’s α, the reliability at the time of development was 0.89 and 0.938 during the study.

### 2.3. Data Collection and Ethical Considerations

Data collection was carried out from 20–27 October 2019. The subjects read the research description and agreed to fill out the questionnaire, seal it, and drop it in the collection box, which took about 15–20 min to accomplish.

This study was conducted with the approval of the Research Ethics Review Committee for the ethical protection of research subjects (CNUH-2019-268). Its purpose, methods, and clauses of anonymity and confidentiality were explained to the participants. All subjects provided written informed consent and participated in the study voluntarily. They were also informed that they could request to stop participating at any point in the study. The data collected from the survey was stored in a secure location and properly disposed of immediately after the study’s completion. Data from a relevant doctoral thesis [17] was used with the consent of the authors.

### 2.4. Statistical Analysis

The collected data were analyzed using the SPSS Statistics 25.0 (IBM Corp., IL, USA) program. The statistical significance was established based on a two-sided test of 0.05, and Cronbach’s α was used to assess the questionnaire’s reliability. The general characteristics were analyzed using frequency, percentage, mean, standard deviation, skewness, and kurtosis. In testing the variables’ normality, skewness, kurtosis, and the Kolmogorov–Smirnov test were used. The t-test, χ^2^ test, Fisher’s exact test, and Scheffe test were performed as post hoc tests to examine the differences in resilience. Moreover, correlation analysis and stepwise multiple regression analysis were performed to investigate the relationship and influence between variables. In particular, the stepwise regression analysis process of this study was performed using variables that are generally judged to be statistically significant according to the method proposed by Eforymson [18]. Variables composed of nominal scales were analyzed after changing them to dummy variables.

## 3. Results

### 3.1. Resilience According to General (Demographic, Work-related) Characteristics

When comparing the differences in resilience according to the demographic characteristic, the significant variables were gender (*t* = 2.232, *p* < 0.05), perceived health status (*F* = 4.724, *p* < 0.05), and reasons for choosing nursing (*F* = 6.598, *p* < 0.01). Furthermore, after comparing the differences in resilience based on work-related attributes, it was found that the subject’s satisfaction of the current ward (*t* = 3.372, *p* < 0.001), satisfaction with clinical practice as a student (*F* = 4.345, *p* < 0.05), and satisfaction with the nursing field (*t* = 20.042, *p* < 0.001) were significant variables (Table 1). On gender, skewness and kurtosis were not normally distributed with an absolute value of 2 or higher, so log transformation was used to confirm the normal distribution.

### 3.2. Subjects’ Nursing Work Environment and Resilience

Table 2 shows the subjects’ nursing work environment and resilience. One sub-variable under the “work environment” variable, “nursing foundation for quality of care”, had resulting skewness and kurtosis that were not normally distributed with an absolute value of 2 or more, so the log transformation was performed to confirm the normal distribution.

### 3.3. Relationship between Nursing Work Environment and Resilience

A positive correlation between work environment and resilience was found (*r* = 0.356, *p* < 0.001) based on the analysis results between the nursing work environment and resilience. This correlation was found while controlling for statistically significant among general characteristics, gender, perceived health status, reasons for choosing nursing, satisfaction with the current ward, and satisfaction with clinical practice as a student.

### 3.4. Factors Affecting Resilience

After controlling for statistically significant variables in general characteristics, hierarchical regression analysis was performed to investigate the effect of the work environment on new nurses’ resilience. In model 1, demographic characteristics such as gender, perceived health status, and reasons for choosing nursing were included in determining their effects on resilience. Model 2 used work-related attributes such as satisfaction with the current ward, satisfaction with clinical practice as a student, and satisfaction with nursing. In model 3, the work environment was added as an independent variable for each area to determine whether the nursing work environment affected the resilience of new nurses even after controlling for exogenous variables.

The analysis results of each model, model 1 (F = 6.836, *p* < 0.001), model 2 (F = 7.9443, *p* < 0.001), and model 3 (F = 9.371, *p* < 0.001), revealed that the regression model was suitable. In addition, the following coefficients of determination, R^2^ = 0.107, R^2^ = 0.198, and R^2^ = 0.338 were found for models 1, 2, and 3, respectively. Moreover, the amount of change in R^2^ increased by 0.091 (*p* < 0.001) and 0.140 (*p* < 0.001) in succeeding models. After inputting the control variable, it was seen that the independent variable is statistically significant in explaining the dependent variable. Models 1, 2, and 3 all had tolerance (TOL) above 0.1 and variance inflation factors (VIF) below 10, confirming that there was no problem of multicollinearity between variables.

In model 3, the sub-variables “nursing foundation for quality of care” and “nurse manager ability, leadership, and support of nurses” were *t* = 3.532 (*p* = 0.001) and *t* = 2.278 (*p* = 0.024), respectively, indicating significance in the positive (+) direction. In contrast, the sub-variable “nurse participation in hospital affairs” was significant in the negative (−) direction (t = –2.087, *p* < 0.038).

Based on the results of the variables’ relative influence on improving the resilience of new nurses, the sub-variables “nursing foundation for quality of care” (β = 0.331), “nurse participation in hospital affairs” (β = –0.211), “nurse manager ability, leadership, and support of nurses” (β = 0.187), “collegial nurse-physician relations” (β = 0.104), and “staffing and resource adequacy” (β = –0.060) were found to affect resilience. Overall, the study’s regression model shows that the work environment impacts the resilience of new nurses by 30.2%. The specific results are shown in Table 3.

## 4. Discussion

This study investigated how the nurses’ work environment affects new nurses’ resilience through hierarchical regression analysis. Statistical analysis results found that the work environment had a significant positive effect on the resilience of new nurses.

On the difference in resilience according to general characteristics, it was found that there was a statistically significant difference in the resilience of new nurses based on their gender, perceived health status, and reasons for choosing the nursing field. In addition, nurses who were assigned to their desired workplace or ward were found to be more resilient than the nurses who were not. However, the difference was not statistically significant. Jung and Park [10] found that the resilience of nurses who wanted to work in the emergency room was significantly higher statistically. Similarly, those who continuously wanted to work in the emergency room were more resilient than those who did not want to work there. This phenomenon implies that nurses who were satisfied with their practice experiences as students and those who chose to go into the nursing field as a vocation are more resilient than their counterparts. In addition, these results can be used as a basis for the fact that preference is an important criterion when assigning new nurses to a working ward in the future.

In contrast, Ying et al. [11] found that intensive care unit nurses over 30 years old showed a significantly higher resilience compared to the 25 and younger group, but not in terms of gender and marital status. No differences were found in resilience in relation to nursing education, working unit, nursing experience, or working experience in the unit. In this study, because the floor ward of new nurses was not limited to 35.6%, and the average age was 23.61 years, the study population, the results are not comparable with that of Ying et al. [11]. Therefore, it is worth considering the work experience of nurses and placing them in their desired ward to increase resilience as a strategy to reduce the turnover intention of nurses. This process can become a strategy for new nurses to increase their satisfaction and ultimately reduce turnover.

This study also examined the scores for each work environment sub-variable and resilience. The sub-variables “nurse manager ability, leadership, and support of nurses” showed the highest average score, while “staffing and resource adequacy”, the lowest score in the study. Kim and Kim [12] found that the sub-variables “nursing foundations for quality of care” had the highest score, followed by nursing participation in hospital affairs. “Nurse manager ability, leadership, and support from nurses”, “staffing and resource adequacy” and “collegial nurse–physician relations” were found in the following order. It is important to note that the points are an integrated score that does not consider the number of items.

When evaluating with consideration for the number of items, their results are similar to those of this study in that “nursing foundation for quality of care” and “nurse manager ability, leadership, and support of nurses” scored the highest. However, the “collegial nurse–physician relations” was the lowest-scoring item in this study. This result is somewhat different in the average number of working months of subjects of Kim and Kim’s [12] was 5.2 months. Therefore, it is necessary to consider the length of time new nurses have worked in a certain workplace or ward by dividing the period in more detail rather than comprehensively evaluating the new nurses who have worked for less than 12 months.

Jung and Park [10] found that resiliency had the greatest influence on post-traumatic growth, while the nursing work environment was not statistically significant. With their finding, the resulting regression model in this study shows that post-traumatic growth is highly related to the turnover of emergency room nurses and that resilience is a very significant variable to such growth. Nonetheless, it does not explain the link between the work environment and resilience. Ying et al. [11] observed that the turnover intention was significantly influenced by marital status, resilience, nursing practice environment. Although this finding suggests that resilience and nursing work environment are significant variables for turnover, the relationship between resilience and work environment was not elucidated either.

Kim and Kim [12] noted that the following sub-variables were influenced resilience, including “staffing and resource adequacy”, “collegial nurse–physician relations”, “nurse manager ability, leadership, and support of nurses”, “nursing foundation for quality of care”, and “nurse participation in hospital affairs”. In particular, after controlling for exogenous variables, the work environment explained the 38% transition shock that was observed. This finding suggests new nurses’ transition shock is greatly affected by the work environment, similar to this study’s results. However, the sub-variable “nursing foundation for quality of care” was the strongest influencing factor in this study, unlike in theirs where “staffing and resource adequacy” was found to be the most influential. The difference is believed to be caused by the ratio of new nurses in the working ward or the number of working months, which implies the necessity to research with more new nurses.

This study presented the need to develop a strategy to improve the working environment and increase the resilience of new nurses by proving that the working environment has a significant influence on the nurses’ resilience. While successful in identifying influential factors, the study was limited by the following. First, the results are based on self-reported scores, implying that subjectivity may be involved in the work environment experienced by the individual. Second, it is difficult to generalize the results of this study because this study did not consider interfering factors such as work places that could affect nurse’s resilience. Therefore, it is thought that additional research considering the work place is necessary.

## 5. Conclusions

This study confirmed that the work environment of new nurses is an essential variable in understanding and explaining their resilience. In particular, it confirmed that the “nursing foundation for quality of care” was the most influential of the work environment sub-variables. The influencing factors identified in this study have an increased impact on resilience based on the detailed aspects of the new nurses’ work environment. These factors could lower the nurses’ turnover intention. Further study using the baseline data collected here is recommended to elucidate the relationship between the work environment and new nurses’ resilience.

As the concept of a work environment changes with time and circumstances, existing cross-sectionals are limited. Therefore, it is necessary to attempt a longitudinal study to measure the degrees of change in the hospital environment and identify the variables that affect resilience over time. Moreover, further research is needed to verify various sub-variables within the work environment. There is insufficient research about developing programs that could improve the work environment, increase the resilience of new nurses in domestic and foreign environments, and verify their effects. Furthermore, it is necessary to study qualitative approaches to understand the changes in new nurses’ resilience based on their changing work environment.

## Figures and Tables

**Table 1 ijerph-19-05158-t001:** Resilience according to the general characteristics (*n* = 233).

**Demographics**						
**Characteristics**		***n* (%)**	**Skewness**	**Kurtosis**	**Resilience**	
					**M + SD**	**T or F (*p*) Scheffe**
Gender	Female	213 (91.4)	−2.976	6.917	3.31 ± 0.52	2.232 * (0.021)
	Male	20 (8.6)			3.59 ± 0.54	
Age (years)	21–22	42 (18.0)	−0.182	−0.899	3.28 ± 0.50	1.199 (0.303)
	23	118 (50.6)			3.30 ± 0.53	
	≥ 24	73 (31.4)			3.41 ± 0.55	
Practicing religion	Yes	76 (32.6)	−0.746	−1.456	3.37 ± 0.60	0.759 (0.449)
	No	157 (67.4)			3.31 ± 0.49	
Cohabitation	With Family	173 (74.2)	1.222	−0.442	3.33 ± 0.53	1.546 (0.215)
	Others	8 (3.4)			3.02 ± 0.42	
	Alone	52 (22.3)			3.37 ± 0.54	
Practiced as a student or intern at current workplace	Yes	197 (84.5)	1.924	1.717	3.33 ± 0.53	−0.234 (0.815)
	No	36 (15.5)			3.35 ± 0.53	
Perceived health status	Healthy (a)	92 (39.5)	0.303	−0.638	3.44 ± 0.62	4.724 * (0.010) a > c
	Usually (b)	123 (52.8)			3.28 ± 0.45	
	Weak (c)	18 (7.7)			3.08 ± 0.42	
Reason for choosing nursing	Easy to get a job (a)	116 (49.8)	0.259	−1.787	3.24 ± 0.50	6.598 ** (0.002) b > a,c
	Vocation (b)	31 (13.3)			3.61 ± 0.47	
	Others (c)	86 (36.9)			3.35 ± 0.56	
With economic obligations	Yes	155 (66.5)	0.705	−1.516	3.31 ± 0.55	−0.886 (0.376)
	No	78 (33.6)			3.37 ± 0.50	
**Work-Related Attributes**						
**Characteristics**		***n* (%)**	**Skewness**	**Kurtosis**	**Resilience**	
					**M + SD**	**T or F (*p*) Scheffe**
Workplace/Ward	Medicine Ward	60 (25.8)	−0.069	−1.225	3.32 ± 0.62	0.255 (0.858)
	Surgical Ward	50 (21.5)			3.31 ± 0.43	
	Intensive Care Unit	83 (35.6)			3.32 ± 0.51	
	Others	40 (17.2)			3.40 ± 0.55	
Work experience in current workplace/ward (months)	< 5	61 (26.2)	0.167	−1.252	3.31 ± 0.55	0.755 (0.556)
	5 ≤ x < 6	52 (22.3)			3.26 ± 0.51	
	6 ≤ x < 7	42 (18.0)			3.40 ± 0.44	
	7 ≤ x < 8	57 (24.5)			3.31 ± 0.64	
	8 ≤ x < 12	21 (9.0)			3.46 ± 0.34	
Orientation period (weeks)	≤7	31 (13.3)	−0.294	−0.779	3.23 ± 0.44	0.643 (0.526)
	8	118 (50.6)			3.36 ± 0.53	
	9–20	84 (36.1)			3.33 ± 0.56	
Assigned to the desired ward	Yes	127 (54.5)	0.182	−1.984	3.39 ± 0.52	1.919 (0.056)
	No	106 (45.5)			3.26 ± 0.54	
Satisfaction with current ward	Yes	177 (76.0)	1.223	−0.508	3.40 ± 0.53	3.372 *** (<0.001)
	No	56 (24.0)			3.13 ± 0.49	
Has breaks on desired days	Yes	119 (51.1)	0.043	−2.016	3.29 ± 0.54	−1.151 (0.251)
	No	114 (48.9)			3.37 ± 0.52	
Experienced turnover	Yes	37 (15.9)	−1.879	1.545	3.47 ± 0.57	1.689 (0.093)
	No	196 (84.1)			3.31 ± 0.52	
Satisfaction with clinical practice as a student	Upper (a)	67 (28.8)	−0.376	0.973	3.49 ± 0.58	4.345 * (0.014) a > b
	Middle (b)	144 (61.8)			3.28 ± 0.47	
	Lower (c)	22 (9.4)			3.20 ± 0.64	
Relation perceptor	Upper	165 (28.8)	−0.573	0.825	3.38 ± 0.54	2.801 (0.063)
	Middle	61 (26.2)			3.23 ± 0.49	
	Lower	7 (3.0)			3.06 ± 0.61	
Satisfaction with the field of nursing	Upper (a)	80 (34.3)	−0.180	0.234	3.49 ± 0.56	20.042 *** (<0.001) c < a,b
	Middle (b)	116 (49.8)			3.36 ± 0.46	
	Lower (c)	37 (15.9)			2.88 ± 0.41	

* *p* < 0.05, ** *p* < 0.01, *** *p* < 0.001.

**Table 2 ijerph-19-05158-t002:** Subject’s nursing work environment and resilience (*n* = 233).

Variable	Total Mean ± SD	Average Mean ± SD	Min–Max	Skewness	Kurtosis
**Work Environment**	79.27 ± 11.77	2.73 ± 0.41	1.03–3.97	−0.113	1.412
Nurse participation in hospital affairs (9 items) *	23.35 ± 4.44	2.59 ± 0.49	1.00–4.00	−0.040	0.324
Nursing foundation for quality of care (9 items) *	26.27 ± 3.69	2.92 ± 0.41	1.00–4.00	−0.329	2.372
Nurse manager ability, leadership, and support of nurses (4 items) *	11.91 ± 1.91	2.98 ± 0.48	1.25–4.00	−0.296	0.695
Staffing and resource adequacy (4 items) *	9.16 ± 2.44	2.29 ± 0.61	1.00–3.75	0.105	−0.712
Collegial nurse–physician relations (3 items) *	8.58 ± 1.51	2.86 ± 0.50	1.00–4.00	−0.424	1.426
**Resilience**	83.09 ± 13.30	3.33 ± 0.53	1.25–4.92	0.119	1.142
Hardness	28.33 ± 5.35	3.15 ± 0.59	1.22–5.00	0.222	0.818
Persistence	27.83 ± 4.68	3.48 ± 0.59	1.00–5.00	−0.031	1.080
Optimism	13.25 ± 2.74	3.31 ± 0.68	1.00–5.00	0.098	0.432
Support	7.74 ± 1.61	3.87 ± 0.80	1.00–5.00	−0.556	0.133
Spiritual in nature	5.94 ± 1.43	2.97 ± 0.71	1.00–5.00	0.159	0.086

* Variables from the Korean Version of Practice Environment Scale of Nursing Work Index (K-PES-NWI).

**Table 3 ijerph-19-05158-t003:** Factors affecting nurses’ resilience (*n* = 233).

Variable	Model 1				Model 2				Model 3			
	B	SE	β	t (*p*)	B	SE	β	t (*p*)	B	SE	β	t (*p*)
Constant	3.851	0.148		25.935 (0.000)	4.133	0.211		19.632 (0.000)	2.615	0.324		8.080 (0.000)
**Demographics**												
Gender	−0.350	0.173	−0.128	−2.029 (0.044)	−0.359	0.166	−0.132	−2.159 (0.032)	−0.337	0.154	−0.124	−2.181 (0.030)
Perceived health	−0.153	0.055	−0.175	−2.752 (0.006)	−0.091	0.055	−0.105	−1.645 (0.101)	−0.097	0.051	−0.112	−1.898 (0.059)
Reason for choosing nursing: Easy to get a job	−0.143	0.072	−0.135	−1.969 (0.050)	−0.124	0.070	−0.117	−1.788 (0.075)	−0.131	0.064	−0.123	−2.036 (0.043)
Reason for choosing nursing: Vocation	0.219	0.107	0.140	2.050 (0.042)	0.175	0.104	0.113	1.695 (0.091)	0.138	0.096	0.089	1.442 (0.151)
**Work-related** **attributes**												
Satisfied with workplace/assigned ward					0.067	0.083	0.054	0.800 (0.425)	0.024	0.077	0.020	0.316 (0.752)
Satisfaction with clinical practice as a student					−0.027	0.060	−0.030	−0.450 (0.654)	−0.013	0.057	−0.015	−0.230 (0.819)
Satisfaction with the field of nursing					−0.212	0.055	−0.274	−3.876 (0.000)	−0.161	0.052	−0.208	−3.069 (0.002)
**Work environment**												
Nurse participation in hospital affairs									−0.227	0.109	−0.211	−2.087 (0.038)
Nursing foundation for quality care									1.144	0.324	0.331	3.532 (0.001)
Nurse manager ability, leadership, and support of nurses									0.208	0.091	0.187	2.278 (0.024)
Staffing and resource adequacy									−0.053	0.060	−0.061	−0.894 (0.372)
Collegial nurse–physician relations									0.110	0.081	0.104	1.352 (0.178)
F (*p*)	6.836 (<0.001)				7.944 (<0.001)				9.371 (<0.001)			
R^2^	0.107				0.198				0.338			
Adjusted R^2^	0.091				0.173				0.302			

Note: The reference data used were males who were not satisfied with their current workplace/assigned ward.

## Data Availability

Data available within the article.

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
