# Peer review of "Factors Affecting the Resilience of New Nurses in Their Working Environment"

_ijerph, 2022, doi:10.3390/ijerph19095158_

Round 1
Reviewer 1 Report
Please:
Explain more about the sampling method and data gathering in the university hospitals. Was simple random sampling, stratified, or systematic? Or was it a non-random sampling method?
High-stress environments, such as an emergency department, ICU, or CCU, will have a different effect on new nurses’ resilience than less-stress environments, such as the inpatient or childbirth wards. Workplace stress level is a confounding factor that was not considered or reported in this study. It may be better to measure the studied factors separately for high-stress and low-stress departments.
Before using an abbreviation, at first write the full form of the phrase. What do you mean by “ER” in the discussion section?
The authors wrote “ because this study did not invest gate the entire population of new nurses” as a limitation. I think this is not a limitation, this problem can be solved by proper and accurate sampling.
Author Response
Dear Editor and Reviewer:
On behalf of my co-autuors, I thank you and the reviewers for your thoughtful comments and recommendations. We have made the following changes to the manuscript to address your concerns. The revisions are shown in RED in the text.
File attached

Reviewer 2 Report
Dear Authors,
The abstract methodological chapter is incomplete and should be sought to display the type, location, time, target group, element number, inclusion and exclusion criterion, method of data collection, main issues, method of statistical analysis, p value . Include numerical results and the value of p.
Author Response
Dear Editor and Reviewer:
On behalf of my co-autuors, I thank you and the reviewers for your thoughtful comments and recommendations. We have made the following changes to the manuscript to address your concerns. The revisions are shown in RED in the text.
File attatched.

Reviewer 3 Report
In this article authors identify the factors of the nursing working environment affecting new nurses ’resilience. The research is interesting, the results obtained indicate factors that may have an impact on the resilience of new nurses.
The discussion section requires some improvement.
- References in text do not correspond to the list of references
- In the discussion, there is no need to repeat the results of own research, if these data are included in the results section. There is also no need to quote the detailed results of research by other authors, it is enough to provide general conclusions regarding the cited research and to engage in discussions with the results of own research.
The discussion needs to be expanded and improvement.
Author Response

(The authors gave the same response as above.)

Round 2
Reviewer 1 Report
The authors have considered the comments, but the study does not have the ability to generalize the results. Moreover, the interfering factors are not controlled.
Author Response
Attached file.
